# Coded Excitation for Ultrasonic Testing: A Review

**DOI:** 10.3390/s24072167

**Published:** 2024-03-28

**Authors:** Chenxin Weng, Xu Gu, Haoran Jin

**Affiliations:** The State Key Laboratory of Fluid Power and Mechatronic Systems, College of Mechanical Engineering, Zhejiang University, Hangzhou 310027, China; 12125027@zju.edu.cn (C.W.); guxu726@zju.edu.cn (X.G.)

**Keywords:** ultrasonic testing, coded excitation, non-destructive evaluation, signal-to-noise ratio, efficient detection

## Abstract

Originating in the early 20th century, ultrasonic testing has found increasingly extensive applications in medicine, industry, and materials science. Achieving both a high signal-to-noise ratio and high efficiency is crucial in ultrasonic testing. The former means an increase in imaging clarity as well as the detection depth, while the latter facilitates a faster refresh of the image. It is difficult to balance these two indicators with a conventional short pulse to excite the probe, so in general handling methods, these two factors have a trade-off. To solve the above problems, coded excitation (CE) can increase the pulse duration and offers great potential to improve the signal-to-noise ratio with equivalent or even higher efficiency. In this paper, we first review the fundamentals of CE, including signal modulation, signal transmission, signal reception, pulse compression, and optimization methods. Then, we introduce the application of CE in different areas of ultrasonic testing, with a focus on industrial bulk wave single-probe detection, industrial guided wave detection, industrial bulk wave phased array detection, and medical phased array imaging. Finally, we point out the advantages as well as a few future directions of CE.

## 1. Introduction

Non-destructive testing is generally defined as follows: under the premise of not causing damage to the specimen, using physical or chemical methods, with the help of devices, to examine the structure, properties, and condition of the interior and surface of the specimen. Industrial ultrasonic testing originated in the early 20th century and is one of the non-destructive testing methods. It uses sound waves with frequencies higher than the range of human hearing, usually referring to sounds with frequencies exceeding 20,000 Hz [1,2,3]. The basic principle is to incident ultrasonic waves into the measured material and capture the reflected sound wave signal through the receiver. By analyzing the characteristics of the signal, such as amplitude, time delay, and frequency, non-destructive testing personnel can detect defects inside industrial materials, such as cracks, corrosion, and voids [4,5,6,7,8]. At first, ultrasonic testing was limited to single-probe bulk wave detection, and then with the advancement of technology, guided wave detection and phased array detection were developed to meet the needs of different fields and applications [9,10,11].

While the principles of medical ultrasonic testing closely resemble those of industrial ultrasonic testing, the probes used are slightly different in structure, and the objects being tested are also converted from industrial materials to human bodies in medical ultrasound [12,13,14]. The rapid development of medical ultrasound can be attributed to the advantages of harmlessness, high sensitivity, and cost-effectiveness. This growth gained momentum with the introduction of the first commercially available ultrasound instrument in the 1960s [15,16]. To help doctors enable the swift identification of patient lesions, medical ultrasound distinguishes itself by emphasizing two-dimensional imaging, presenting information about a cross-section of the body through detection modes like sector scans with the phased array probe. Due to the dynamic nature of some detection objects, such as the heart and blood flow, high frame rate imaging is required, which can quickly refresh the image [17,18,19].

The signal-to-noise ratio (SNR) and imaging frame rate are two important indicators in both industrial ultrasound and medical ultrasound, meaning clearer and faster, respectively [20,21,22,23]. Conventional ultrasonic testing emits short pulses. Due to the low incident energy, it can result in an inadequate SNR when detecting materials with high acoustic attenuation or deep tissues inside the human body. When you want to increase the detection signal-to-noise ratio, the first method is to increase the excitation voltage of the pulse, but this will affect the probe life, and the use of high voltage pulses in medical ultrasonic testing is prone to cause harm to the human body. The second method is to increase the number of excited oscillations, but in doing so, the resolution will be reduced, meaning that the ultrasound’s ability to distinguish small defects and tissues will be reduced. The third method is using an averaging algorithm to improve the SNR, but doing so means multiple transmissions and receptions, which can seriously affect detection efficiency. In addition to the SNR, if you want to improve the imaging efficiency of conventional ultrasonic testing, you generally have to reduce the number of ultrasound transmissions, which in turn reduces the SNR, and the third method described above will not work. It can be seen that SNR, efficiency, and resolution are coupled together. Therefore, the question of how to improve the SNR and imaging frame rate while guaranteeing the resolution becomes expected. Fortunately, coded excitation (CE) can be a promising solution [24,25].

CE originated in the field of radar and has been used in radar for more than 50 years [26,27]. Its principle can be simply summarized as follows: the transducer transmits a modulated long pulse and recovers the signal for pulse compression. Takeuchi was the first researcher to consider the application of coded signals to ultrasound imaging in 1979 [28], and since then, CE has been increasingly researched in the field of ultrasonic detection, and the ability of CE to improve the SNR and frame rate has gradually matured [29,30]. CE generally emits frequency-modulated or phase-modulated longer signals to increase the emitted energy, and some common codes are used for CE, such as chirp, Barker, and Golay [31,32,33]. After receiving the signal, a decoding filter is employed for pulse compression to maintain the original axial resolution and enhance the SNR. The most common method of pulse compression is matched filtering, but mismatch filtering and inverse filtering with different windows have also been proposed to suppress the distance lobes. As for the specific principles of CE, they will be introduced specifically in Section 2.

Due to the numerous advantages of CE, it has been widely used in the field of ultrasonic testing. CE was first used in the medical ultrasonic testing field for improving the imaging signal-to-noise ratio, the imaging depth, and the imaging frame rate and was used to detect objects such as cysts located deeper in the body, as well as transcranial, heart, and blood flow [34,35,36,37,38,39]. Since CE has many advantages, it is developing rapidly in medical ultrasound, and many medical ultrasound machines nowadays have integrated CE functions to improve imaging, which is quite helpful to improve the correctness of the doctor’s diagnosis. Witnessing the successful application of CE in medical ultrasonic testing, researchers in the field of industrial ultrasonic testing have also begun to assimilate CE [40,41,42,43]. Initially, CE was used for industrial bulk wave single-probe inspection for some simple industrial materials, and the function was not limited to defect detection but also for material thickness measurement. Next, CE was introduced in the field of guided wave for the inspection of bar-shaped as well as flat materials. With the development of technology, new forms of air-coupled ultrasonic testing and phased array ultrasonic testing have emerged, in which CE also has a place, and spatial coding will be introduced here (note: CE contains both temporal and spatial coding but generally only refers to temporal coding). As for a more detailed introduction to the application of CE in ultrasonic testing, it will be discussed in Section 3.

This paper reviews coded excitation in the field of ultrasonic testing from the following aspects. Firstly, in Section 2, the basic principles of ultrasonic detection will be briefly introduced, followed by highlighting the improvements made by CE in them. We will then briefly deduce its principles based on its origins in radar. Then, we will introduce some common forms of coding such as the chirp signal. How to optimize CE is equally important, so we will also introduce many ways to optimize CE. In Section 3, the improvement in testing efficiency by CE and the improvement in spatial coding for SNR, both of which are refreshing and effective uses, will be presented, although not as many applications as classical CE. In Section 4, research and application examples of CE in industrial ultrasonic testing and medical ultrasonic testing will be provided to discuss the advantages and limitations of CE in practical applications. In Section 5, a comprehensive table detailing the various types of coded excitation and their characteristics will be presented, and the future research directions for coded excitation will be summarized and discussed. 

## 2. Principles and Optimization of Coded Excitation Ultrasonic Testing

### 2.1. Basic Principles of Ultrasonic Testing

Ultrasonic testing (UT) is a commonly used method for detecting internal defects in materials and internal tissues in the human body. It can generally be divided into five steps: emitting ultrasonic waves, propagating ultrasonic waves, receiving and recording data, signal processing and analyzing, and evaluating the results, as shown in Figure 1.

Emitting ultrasonic waves. UT utilizes a transducer to generate ultrasound waves, which are then propagated into the tested object. Generally, the frequency range of ultrasound used in industrial UT is between 0.5 and 10 MHz, and the frequency range used in medical UT is between 2 and 15 MHz. There are also different types of transducers, such as single-element transducers, phased array transducers, etc.Propagating ultrasonic waves. Ultrasonic waves propagate in the tested object and interact with the interface or defects in the material, resulting in reflection and scattering, and the wave after the interaction is the useful signal to be captured.Receiving and recording data. A probe is used to receive reflected or scattered ultrasonic waves and convert them into electrical signals. These electrical signals are recorded for further analysis.Signal processing and analyzing. The received signals are processed and analyzed, including the amplification and filtering of the electrical signals, as well as the analysis of the amplitude, time delay, frequency, and other characteristics of the ultrasonic signals. By analyzing the extracted useful information, it is possible to determine the dimensions of the material, the characteristics of defects, or human tissue.Evaluating the results. Based on the results of the analysis, the integrity and reliability of the tested material and whether the body tissue is diseased are assessed. Based on the results of the assessment, further measures can be taken, such as repairing and replacing the material and developing a treatment plan for the patient.

CE can be regarded as an improvement in the first and fourth steps of the above five steps, as shown in Figure 1. In the step of emitting ultrasound, CE replaces the traditional short pulse with a modulated long pulse with a longer duration [44]. Therefore, it can increase the total energy emitted without increasing the peak emission amplitude. In the step of signal processing and analyzing, since long pulses are transmitted, long echo signals are also received. In order to maintain a resolution similar to that of a conventional short pulse, after the signal has been received, the echo signal needs to be pulse compressed (or decoded) using a filter that converts the long time-scale pulse into a narrow pulse with a high peak power.

Compared with traditional short-pulse excitation, CE utilizes the above methods to improve the detection SNR and increase the detection depth without increasing the peak emission amplitude and maintains the resolution while improving the detection capability.

### 2.2. Basic Principles and Classification of Coded Excitation

#### 2.2.1. Using Matched Filters to Achieve the Highest Signal-to-Noise Ratio

When the ultrasonic probe receives the echo signal, it inevitably receives noise, mainly electronic noise (Gaussian white noise) [45]. In order to improve the ability of ultrasonic testing, it is necessary to suppress noise and interference when processing echo signals, and to highlight useful signals, using filters is a good solution. Figure 2 is the flowchart of ultrasound echo signal processing, where s(t) is the noise-free echo signal, N(t) is Gaussian white noise with a mean of 0, and the bilateral power spectral density is $\frac{N_{0}}{2}$; r(t) is the signal to be processed, H(ω) is the filter frequency response, and y(t) is the output signal.

If the maximum output signal-to-noise ratio criterion is adopted, the matched filter is the corresponding filter [46]. The derivation of the matched filter is relatively complicated, and the final conclusion will be given here directly. For specific derivation, please refer to Fundamentals of Radar Signal Processing. If the SNR is defined as the ratio of the instantaneous power corresponding to the maximum value of the useful signal *S* to the average power of the noise *N*, the SNR of the echo after passing through a filter can be derived as follows:(1)$$\frac{S}{N}\leq \frac{2E}{n_{0}}$$
when the frequency response of the filter meets the following:(2)$$H(\omega )=kS^{*}(\omega )e^{-j\omega t_{0}}$$

In this case, the above equation becomes an equation (k is a constant not equal to zero), at which time the SNR of the output reaches its maximum value:(3)$${\left(\frac{S}{N}\right)}_{\max}=\frac{2E}{n_{0}}$$

This shows that the output SNR of a linear system is maximum when its frequency response function *H*(*ω*) is the complex conjugate of the input signal, and this linear system is called matched filter. Its output SNR can reach the maximum, which is equal to twice the echo signal energy 2E divided by the noise power $n_{0}$ of the unit bandwidth.

#### 2.2.2. Frequency Modulation Coded Waveform

First of all, the traditional emission method cannot continue to be used due to its significant drawbacks. The traditional testing method transmits a simple pulse with only two parameters, amplitude and pulse width, as shown in Figure 3a. The use of such pulses will lead to the coupling of the signal-to-noise ratio and the range resolution, because to increase the SNR, it is necessary to increase the pulse width to improve the incident energy, and the increase in the pulse width will reduce the range resolution. At the same time, the simple pulse cannot be used with matched filter, although the matched filter can achieve the maximum output signal-to-noise ratio; the output of the simple pulse is a triangular waveform as shown in Figure 3b, and the echoes of two nearby targets can be easily mixed together. Therefore, a number of special waveforms are used to accomplish the task, which are collectively referred to as coded waveforms.

Linear frequency modulation (LFM) signal is the most commonly used coded waveform, which is also called chirp signal [47,48]. It is defined as follows:(4)$$x(t)=\cos(\pi \frac{\beta }{\tau }t^{2}),0\leq t\leq \tau$$

Figure 4 gives the corresponding waveform x(t), where β is the bandwidth and τ is the pulse width, expressed in complex form as follows:(5)$$x(t)=e^{j\pi \beta t^{2}/\tau }=e^{j\theta (t)},0\leq t\leq \tau$$

The instantaneous frequency of this waveform is the integral of the phase function. Generally, β>0:(6)$$F_{i}(t)=\frac{1}{2\pi }\frac{d\theta (t)}{dt}=\frac{\beta }{\tau }t\;\mathrm{Hz}$$

Figure 5 gives the matched filter outputs of two LFM waveforms with a βτ of 10 and 100, respectively. As for the βτ, it is an indicator to measure the coded waveform. Generally speaking, the larger the βτ, the better the performance of the encoded waveform. It can be seen that the matched filter output of the LFM waveform is much narrower compared to the simple pulse, which means that the resolution is much higher, and the larger the βτ, the higher the resolution. In this way, the SNR and the distance resolution are decoupled; this process is also known as pulse compression [49,50]. However, please note that the output result of LFM will form some sidelobes similar to sinc signals; these sidelobes will affect the detection quality, and we will introduce the optimization scheme to solve this problem later.

#### 2.2.3. Typical Binary Phase Coding (Barker Code, Golay Code)

In addition to LFM, the second main type of coded waveform is the phase-encoded waveform. The most classic is waveform coding based on Barker and Golay sequences, so that the absolute phase is converted between two or more determined values at fixed intervals within the pulse duration. This pulse can be regarded as a collection of N continuous sub-pulses $X_{n}(t)$ with a pulse width of $\tau_{c}$. Each sub-pulse has different phases, expressed as follows:(7)$$x(t)=\sum_{n=0}^{N-1}x_{n}(t-n\tau_{c})$$
(8)$$x_{n}(t)=\left\{\begin{array}{cc} \exp(j\phi_{n}), & 0\leq t\leq \tau_{c}\\ 0, & else \end{array}\right.$$

The total pulse width is $\tau =N\tau_{c}$, the individual sub-pulses are called chips, and in the case of two-phase encoding, there are only two phase states $\phi_{n}$, usually 0 and π.

Barker code

The Barker code is the most commonly used two-phase code, which is a specific set of binary sequences that can make the ratio of the peak output of the matched filter to the highest sidelobe N:1, as shown in Figure 6b; please note that the normalized time on the horizontal axis implies that the specific time unit depends on the actual setting of the chip length. The Barker code was proposed and developed by physicist Robert M. Barker. Table 1 lists all the Barker codes, and one of its disadvantages is its small number, with a maximum of 13 bits [51]. Figure 6a shows the Barker code waveform when n = 13. It can be seen that this waveform is equivalent to modulating a carrier based on the sequence (the time duration of each chip is set to 0.4 μs). When the sequence is “+1”, the chip takes the positive phase, and when the sequence is “−1”, the chip takes the negative phase. The length of each chip is related to the center frequency of the transducer. In Figure 6, the phase conversion can be seen clearly.

The advantage of the Barker code is that it has good autocorrelation, and the output of matched filter has a narrow main lobe and low sidelobe, which can achieve a high SNR and excellent resolution. Moreover, the same location only needs one transmitting and receiving process, which is more convenient.

The biggest disadvantage of the Barker code is the limitation of the code length. Generally speaking, increasing the code length can increase the incident energy to improve the SNR, but the Barker code has only 13 bits at most, which limits its ability of SNR improvement.

Golay code

A Golay code is a kind of sequence with excellent performance, and the most significant feature is that it works in pairs; it was proposed and developed by physicist Marcel J.E. Golay. A pair is composed of two binary sequences, $A[n]=[a_{0},\;a_{1},\;\ldots ,\;a_{N-1}]$ and $B[n]=[b_{0},\;b_{1},\;\ldots ,\;b_{N-1}]$, of the same length N such that $a_{i}$, $b_{i}\in \{-1,+1\}$. Like the Barker code above, the Golay code also needs to be processed by matched filter pulse compression [52,53]. The difference is that it needs to perform matched filter processing on the two sequences, respectively, and then sum the results. As shown in Figure 7, the matched filtering results of A[n] and B[n] have equal sidelobe sizes but opposite signs. After adding the two, an ideal function is obtained by the following:(9)$$C_{AA}[n]+C_{BB}[n]=\left\{\begin{array}{cc} 2N, & n=0\\ 0, & n\neq 0 \end{array}\right.$$
where $C_{AA}[n]\;$and $C_{BB}[n]$ represent the matched filter outputs of A[n] and B[n], that is, the autocorrelation function. It can be found that the final result is that there is a high amplitude response only at the horizontal axis of 0, and the sidelobes at other positions of the horizontal axis are eliminated.

Like the Barker code mentioned above, the discrete sequence cannot directly drive the transducer, so it is necessary to modulate a carrier, which can be modulated in the form of the trigonometric function described above, or the modulation scheme of binary phase-shift keying (BPSK). Each chip is composed of a plurality of positive pulses (+1) and negative pulses (−1). On the one hand, this can be applied to more ultrasonic instruments, and on the other hand, it can also prevent the equipment from being damaged when it is at a single high or low level for too long.

The most significant advantage of Golay codes lies in their ability to eliminate the sidelobes when two Golay codes are overlapped after filtered. This results in the generation of a narrow pulse with a high peak value, which cannot be achieved by other coding methods [54,55]. In fact, the maximum peak value of a Golay code is 2n, twice the code length, further highlighting its superior properties. In addition, Golay codes do not have a limit of code length like Barker codes, and theoretically, it is possible to generate unlimited-length Golay codes. Based on these advantages, Golay codes are theoretically the most ideal coding method.

One notable limitation of Golay codes is that they require the excitation to be transmitted twice, which can significantly impact the efficiency of imaging applications [56,57,58,59,60]. In real-time ultrasonic testing, for example, the echo signals generated by the A and B codes often do not correspond due to the relative movement of the probe position. This can lead to information errors during the decoding process, which can limit the accuracy of the results obtained.

#### 2.2.4. Diverse Alternative Forms of Coding (P3, P4 Code, M-Sequence)

P3, P4 code

The Barker code mentioned above is a type of two-phase coding, while the P3 code and P4 code belong to a type of polyphase coding. In polyphase coding, the chip phase can be encoded with any value, allowing for more flexibility in the coding scheme [61,62]. This can be seen as a specific definition of $\phi_{n}$ for Equation (8):(10)$$P3:\;\phi_{n}=\left\{\begin{array}{ll} \frac{\pi }{N}n^{2}, & n=0,2,\ldots ,N-1\;(N\;\mathrm{is}\;\mathrm{odd})\\ \frac{\pi }{N}n(n+1), & n=0,2,\ldots ,N-1\;(N\;\mathrm{is}\;\mathrm{even}) \end{array}\right.$$
(11)$$P4:\;\phi_{n}=\frac{\pi }{N}n^{2}-\pi n,n=0,1,2,\ldots ,N-1$$

Multi-phase codes have lower sidelobe levels and larger Doppler tolerance than two-phase codes. The matched filter output of a 64 bit P3 code is given in Figure 8. Lower sidelobe levels mean less interference, and Doppler tolerance is a concept in radar, where a higher Doppler tolerance means a higher ability to discriminate between different distances and different targets [63].

M-sequence

The M-sequence, which stands for maximum-length linear-feedback shift register sequence, is created by using both linear-feedback shift registers and heterodyne gate circuits to generate them. The generating polynomial of the M-sequence is the primitive polynomial, and the longest possible period that a binary linear-feedback shift register sequence can generate is $N=2^{n}-1$, where n is the number of stages of the generating polynomial. The specific method of generation is not necessary to be clear; it is a sequence of 1s and 0s, which means that using the above method can transmit waveform modulation as well as decoding [64,65,66]. M-sequences are easy to generate, regular, and have many excellent properties, including the following:(1)Shiftability: the M-sequence obtained by cyclic shift including left shift or right shift is still an M-sequence.(2)Equilibrium: the number of 1s and 0s in the M-series is basically the same, among which 1 is more than 0.(3)Autocorrelation: The M-sequence is known for its excellent autocorrelation properties. As a pseudo-random code, when the M-sequence is shifted by an integer multiple of its period, the autocorrelation reaches its maximum value of 1. For all other shift values, the autocorrelation value is −1/p, where p represents the period of the M-sequence.(4)Strong anti-interference: in communication, an M-sequence can be used for spread spectrum communication.(5)Good steganography: the M-sequence is a pseudo-random sequence, similar to white noise.

In the field of ultrasound, people are mainly interested in the third point mentioned above. As for the “encryption” mentioned in the fourth and fifth points, it has a great role in military communications, and in the field of ultrasound, this “encryption” can be used to achieve the efficient scanning of multi-beam parallel transmission, which will be further introduced later.

### 2.3. Optimizations of Coded Excitation

#### 2.3.1. Optimizations of Post-Processing

Mismatched filtering

Mismatched filtering is a post-processing optimization technique for linear frequency modulation (LFM) waveforms. It can be visualized as applying a windowing function to the matched filter [67,68]. The primary objective of this technique is to minimize the sidelobes in the output, thereby reducing interference. By employing mismatched filtering, the undesired effects of sidelobes can be mitigated, resulting in improved signal quality and reduced interference. The matched filter has been previously described. Its impulse response is denoted as $h_{match}(t)$, and the window function is represented by ω(t). The impulse response of the mismatched filter can be expressed as follows:(12)$$h_{mismatch}(t)=\omega (t)h_{match}(t)$$

ω(t) can be any window function such as the Hamming window, kaiser window, Chebyshev window, and more. Among these window functions, the Chebyshev window is widely regarded as the most commonly used due to its excellent properties. It offers the desirable characteristic of having the smallest main lobe width for a given sidelobe amplitude level. This makes it highly suitable for applications in the radar field, where reducing sidelobes and achieving high resolution are essential requirements.

The outputs of two mismatched filters, weighted by the Hamming window and Chebyshev window, as well as the outputs of the matched filters, are depicted in Figure 9, labeled as a and b, respectively. T represents the time width of the original LFM signal. In the output results using the Hamming window, the first sidelobe near the main lobe is attenuated to approximately −40 dB. Subsequently, employing the Chebyshev window further diminishes the first sidelobe to −60 dB, demonstrating an even greater reduction in sidelobe levels. Indeed, mismatched filtering provides a significant attenuation of sidelobes but at the expense of widening the main lobe, leading to a loss of resolution. This trade-off should be carefully considered when using this method for optimization. It is important to weigh the advantages and disadvantages based on specific application requirements. In some cases, reducing sidelobes may take precedence over maintaining high resolution, while in other scenarios, preserving resolution may be more critical. Balancing these factors is essential to achieve the desired performance and optimize the system accordingly.

Inverse filter

Based on the principle of pulse compression, the ultimate goal is to compress a long-coded transmission pulse into a δ function, which represents an infinitesimally narrow pulse. To achieve this, a common approach is to operate in the frequency domain and utilize a filter to process the spectrum of the coded pulse. By carefully designing and applying the appropriate filter, it is possible to achieve effective pulse compression, resulting in improved range resolution and target detection capabilities in ultrasonic systems [69].

Assuming that the spectrum of the coded pulse x (t) is X (ω), the transfer function H (ω) of the inverse filter is as follows:(13)$$H(\omega )=\frac{1}{X(\omega )}=\frac{X^{*}(\omega )}{X(\omega )X^{*}(\omega )}=\frac{X^{*}(\omega )}{|X(\omega ){|}^{2}}$$

Based on the above formula, the frequency spectrum of the coded pulse is derived using the Discrete-Time Fourier Transform (DTFT), and the impulse response of the inverse filter can be obtained by calculating the reciprocal and performing the inverse transform. In practical applications, the Discrete Fourier Transform (DFT) is commonly utilized in place of the DTFT, and this transformation is often accomplished using the Fast Fourier Transform (FFT), a rapid algorithm for computing the DFT. In the calculation, the coded pulse needs to be properly zeroed before the FFT, and the IFFT result needs to be properly truncated to obtain an inverse filter of appropriate length.

There is also literature mentioning the use of Minimum Mean Square Error (MMSE)-based inverse filtering, which has the same objective as the inverse filter but differs slightly in the process; the specific details of this approach are not elaborated here. For binary-coded sequence waveforms, the sidelobe suppression performance of the MMSE-based inverse filter is generally slightly better than that of the conventional inverse filter.

Wiener filter

Some literature suggests that for coded excitation ultrasonic testing systems, in the presence of both structured noise and Gaussian white noise, the optimal pulse compression filter is the Wiener filter [70,71,72], which can be expressed as follows:(14)$$H(\omega )=\frac{X^{*}(\omega )}{N_{0}+k|X(\omega ){|}^{2}}$$

Among them, X(ω) represents the frequency spectrum of the transmitted encoded pulse x(t), $N_{0}$ represents the power spectral density of Gaussian white noise, and k represents the average scattering intensity of the target. When the SNR is small, the Wiener filter expressed above degenerates into a matched filter; when the SNR is large, it degenerates into an inverse filter.

In practical applications, it is important to fully consider the frequency response of the probe and the properties of the target. Sometimes, using a Wiener filter may achieve better output results.

#### 2.3.2. Optimizations of Waveform Design

Optimize emission waveform based on transducer response

Typically, the resolution after pulse compression is directly related to the bandwidth of the echo signal. When the transmitted signal passes through the ultrasound probe, the probe’s pulse response results in the significant attenuation of the low- and high-frequency components of the transmitted signal, acting as a bandpass filter, thereby limiting the bandwidth of the echo signal. Therefore, compensating for the pulse response of the ultrasound probe to increase the bandwidth of the echo signal is an optimization method for improving resolution [73].

In traditional LFM excitation systems, the transmitted LFM signal s(t) has a constant envelope. However, it is possible to make improvements by introducing an amplitude-weighted pre-distorted LFM encoded signal $S_{AM}(t)$. This can be expressed as follows:(15)$$s_{AM}(t)=A(t)s(t)$$
where A(t) is the amplitude weighting function. Its purpose is to counteract the effect of the pulse response of the transmitting probe on the transmitted signal, allowing for the bandwidth of the echo signal to not be limited by the probe. Ideally, A(t) can completely counteract the probe’s effect on the transmitted signal, resulting in a transmitted signal after passing through the probe that is a constant envelope LFM signal, which can be represented in the frequency domain.
(16)$$F[A(t)s(t)]=\frac{F[s(t)]}{F[g(t)]}$$

After obtaining A(t), the required $s_{AM}(t)$ can be obtained according to Equation 15. In Equation 16, g(t) represents the pulse response of the probe, and F denotes the Fourier transform. Typically, the pulse response of the probe is a cosine wave modulated with a Gaussian envelope. Therefore, A(t) can be expressed as the reciprocal of a Gaussian function:(17)$$A(t)=\exp[{\left(t-T/2\right)}^{2}/D^{2}]$$

In the equation, D is the amplitude weighting coefficient, which represents the degree of amplitude weighting. A smaller value of D indicates a higher degree of amplitude weighting for the LFM transmitted signal, which means that the amplitude of the LFM signal is larger at the low and high frequencies (i.e., the beginning and end). The value of D is related to the pulse response of the probe, the bandwidth of the transmitted signal, and the time duration.

RLFM and its spectrum optimization

When we want to transmit LFM signals to improve the SNR, it is necessary to consider whether the available equipment can transmit signals with continuous amplitude variations. Typically, devices that offer the flexibility to edit the shape of the transmitted waveform come with a higher price. For most devices, they are only capable of transmitting rectangular pulses with either a high or low level. Therefore, as a compromise, the option is to transmit rectangular linear frequency-modulated (RLFM) signals [74,75,76,77], also known as pseudo-chirp signals, as shown in Figure 10.

However, transmitting such alternative signals does come with certain drawbacks. Compared to regular LFM signals, these signals exhibit more oscillations in their frequency spectra. It is known that a flatter and broader frequency spectrum is more advantageous for pulse compression, resulting in narrower main lobes and lower sidelobes. Therefore, there is literature mentioning a recursive optimization method for RLFM signals. As shown in Figure 11, the optimization process involves continuously modifying the RLFM signal by recursively adjusting the width of the rectangular pulses in order to achieve the desired spectral flatness ξ within the bandwidth of interest. The definition of ξ is as follows:(18)$$\xi_{i}=\frac{\sqrt{\int_{f_{\min}}^{f_{\max}}|\bar{\left|S_{i}(f)\right|}-|S_{i}(f)|{|}^{2}}}{\bar{\left|S_{i}(f)\right|}}$$
where ${|S}_{i}\left(f\right)|$ represents the magnitude spectrum at the *i*-th step, and its average is $\bar{{|S}_{i}\left(f\right)|}$. In each recursive step, the spectral flatness is considered as a metric, taking into account its variability or flatness within the bandwidth, and normalized by its average, considering the energy within the bandwidth. The smaller the ξ, the better the flatness. A threshold can be set, and when the flatness ξ is below it, the optimization process is considered to have converged and can obtain an excitation signal that maximizes the flatness of the spectrum with the greatest possible energy. The result of the pulse compression for the optimized signal is shown in Figure 12.

The combination of different encoding methods

The Barker code, mentioned earlier, is widely used due to its excellent performance and convenience. However, it also has some drawbacks. For example, one or more sine signals are typically used as modulation carriers for Barker coding. As a single-frequency signal, the time–bandwidth product of a sinusoidal carrier wave is approximately equal to 1. The bandwidth determines the resolution of detection, while the time duration determines the signal-to-noise ratio of detection, hence there exists a contradiction between the two [78].

Considering the limitation of using sinusoidal signals as carriers, it is possible to apply LFM to the Barker code to achieve a time–bandwidth product greater than 1. This approach is referred to as LFM–Barker encoding excitation [79,80].

When using pulse compression filters for post-processing, it is important to pass through two separate filters, where $h_{1}(t)$ represents the pulse compression filter for the LFM carrier pulse, and $h_{2}(t)$ represents the pulse compression filter for the Barker code. In general, the use of matched filtering is sufficient:(19)$$h(t)=h_{1}(t)*h_{2}(t)$$

The filtered result is shown in Figure 13, indicating that compared to the result of conventional Barker code encoding, the main lobe becomes narrower, signifying higher resolution. Similarly, similar modulation enhancements can be made for Golay encoding with similar principles.

## 3. Coded Excitation for Improving Detection Efficiency and Spatial Encoding

### 3.1. Coded Excitation for Improving Detection Efficiency

The primary and most important application purpose of coded excitation is to significantly improve the SNR and detection penetrative capability. However, it is important to remember that, as mentioned earlier, coded excitation originates from the fields of radar, sonar, and military communication. This necessitates that coded excitation not only improves the SNR but also possesses a certain degree of confidentiality. In other words, different coded signals should appear locked, preventing mutual decoding. This aspect adds the potential for encoding incentives to increase detection efficiency. In ultrasonic detection, it is generally not possible to achieve the simultaneous detection of multiple beams because these beams would interfere with each other. However, the parallel transmission of multiple ultrasonic beams can be achieved through designing and transmitting “mismatched” encoding pulses, i.e., pulses with high autocorrelation and low cross-correlation. This significantly improves detection efficiency without sacrificing the ability to enhance the SNR. Below will be some specific forms of mismatched coded excitations.

Mismatched LFM signal

The most commonly used coded excitation signal is the LFM signal. Multiple LFM signals can be generated by changing their parameters, achieving a mismatched effect with high autocorrelation and low cross-correlation [81,82,83]. The most common approach is to keep the bandwidth and duration of the LFM signal unchanged while varying the sweep rate, i.e., sweeping through the same frequency range at different rates. Figure 14a shows the frequency variation of seven mismatched LFM signals. As long as the slope of the frequency variation is different, the desired mismatch can be achieved. Furthermore, this method allows for the creation of any desired number of mismatched LFM signals.

Figure 14b provides the autocorrelation results of the FM4 signal and the cross-correlation results of the FM2, FM3, and FM5 signals. It can be observed that the autocorrelation results of such signals exhibit a high amplitude and narrow main lobes, while the cross-correlation results show lower-level signals similar to background noise. Therefore, this type of signal is capable of achieving the desired mismatched effect. Furthermore, for mismatched LFM signals, a longer pulse width in the time domain leads to a better mismatch effect.

Orthogonal Golay pairs

As mentioned earlier, the main drawback of Golay coding is the need to transmit and receive twice in order to decode a single scanning signal, resulting in only half the efficiency compared to conventional scanning methods. However, the previous idea can be further explored by using the method of simultaneously transmitting multiple pulses to improve the testing efficiency. In medical synthetic aperture imaging, orthogonal Golay coding can be utilized to enhance testing efficiency [84,85,86]. 

For example, consider a pair of orthogonal Golay codes: A1 [+1, +1, +1, −1] and A2 [+1, −1, +1, +1]. Using the principle of cross-correlation inherent in Golay coding, another pair of orthogonal Golay codes can be designed: B1 [+1, +1, −1, +1] and B2 [+1, −1, −1, −1]. A1 and A2 as well as B1 and B2 are mutually orthogonal, while A1 and B1 as well as A2 and B2 form complementary Golay pairs. Therefore, during one transmission, it is possible to simultaneously transmit the two encoding pulses corresponding to A1 and A2, and in the next transmission, simultaneously transmit the two encoding pulses corresponding to B1 and B2. The mentioned orthogonality ensures that the two simultaneously transmitted pulse echo signals can be separated after performing matched filtering. As a result, the medical synthetic aperture imaging process, which originally required four transmissions, can now be completed with only two transmissions, effectively doubling the efficiency. The transmission and imaging process are illustrated in Figure 15.

Gold sequence

In the previous text, the M-sequence was mentioned. The M-sequence is short for “maximum-length linear-feedback shift register” and is characterized by good autocorrelation performance and low cross-correlation, making it suitable for use as pulse sequences for multi-beam parallel transmission. The generation of an M-sequence can be determined by the characteristic polynomial f(x) of a shift register sequence generator:(20)$$f(x)=c_{0}+c_{1}x+c_{2}x^{2}+\cdots +c_{n}x^{n}=\sum_{i=0}^{n}c_{i}x^{i}(c_{0}=1,c_{n}=1)$$

In the equation, $c_{i}$ represents the feedback coefficients, where ci = 0 indicates that there is no feedback for that position, and $c_{i}=1$ indicates that there is feedback. The feedback coefficients can only take the values “0” or “1”. Although M-sequences exhibit good autocorrelation and cross-correlation properties, the number of available address codes for M-sequences is limited. To address this issue, it is necessary to introduce combinatorial codes, and Gold sequences are an important class of combinatorial codes [87].

By performing modulo-2 addition on the aforementioned M-sequences, Gold sequences are obtained. Similarly, Gold sequences also exhibit good autocorrelation and cross-correlation performance. Figure 16 displays the autocorrelation and cross-correlation results of Gold sequences. It can be observed that after pulse compression through autocorrelation, a very narrow waveform is formed, which is beneficial for target resolution. Additionally, the signal amplitude of the cross-correlation is much lower than the peak value of the autocorrelation, allowing for non-interfering multi-beam parallel transmission.

### 3.2. Hadamard Spatial Encoding

In the previous section, a large amount of coded excitation content was introduced, which generally involves the design of transmitted waveform and post-processing for the signals, commonly referred to as time encoding. Spatial encoding is another form. It is important to note that while both can be called encoded ultrasound, spatial encoding and time encoding differ significantly in principle. In general, the default “encoded excitation” refers to the previously mentioned time encoding. The application of spatial encoding is not as widespread as time encoding, and it has more limitations. However, its purpose is the same as time encoding, which is to significantly increase the SNR.

Hadamard encoding is the most commonly used spatial encoding [88,89]. The use of Hadamard spatial encoding requires a prerequisite condition, which is that the device used must be a phased array ultrasound probe composed of multiple piezoelectric elements, assuming that the number of elements is N.

Firstly, it is necessary to have a basic understanding of Total Focusing Method (TFM) ultrasound imaging. This algorithm is based on full matrix capture (FMC), and if the phased array probe has N elements, a total of N × N transmit and receive operations need to be completed [90,91,92]. This means that the ultrasound is first transmitted by element 1, and all elements receive the signals. Then, the ultrasound is transmitted by element 2, and again all elements receive the signals. This process is repeated until N, resulting in a matrix of N × N data, as shown in Figure 17. 

Like the TFM, Hadamard spatial encoding also requires FMC. For a phased array transducer with N elements, N transmission and reception operations are needed. However, Hadamard spatial encoding applies a special Hadamard encoding sequence during the transmission process.

Next, we will introduce Hadamard spatial encoding step by step.

The first step is to construct a Hadamard matrix. Suppose an N-order square matrix satisfies the following condition:(21)$$H_{N}H_{N}^{t}=NI_{N}$$

The matrix $H_{N}$ is then referred to as the Hadamard matrix. For example, a second-order Hadamard matrix: $\left[\begin{array}{cc} 1 & 1\\ 1 & -1 \end{array}\right]$. In practical applications, the matrix order is often higher and aligned with the number of transducer elements in the array.

The second step is to simultaneously excite N transducer elements using the Hadamard matrix. The coding for the m-th transmission corresponds to the coefficients of the m-th column of the Hadamard matrix $H_{N}$. With the example of the second-order Hadamard matrix given earlier, where the first column contains two 1s, for the first transmission, the two elements of the phased array probe (in reality, there are more elements in a phased array probe, but this is based on the assumption of the matrix order) simultaneously emit positive pulses, and then each element receives the echo. The second column of the matrix contains 1 and −1, so for the second transmission, one element of the probe emits a positive pulse while another emits a negative pulse, and then each element receives the echo; the method is shown in Figure 18; $r_{ij}(t)\;$represents the signal emitted by the i-th element and received by the j-th element, $R_{mn}(t)$ represents the signal received by element n during the m-th emit process, and N(t) represents the noise signal. In this way, $N^{2}$ echo signals are obtained, and this matrix is denoted as W(t).

The final step is decoding, as follows:(22)$$K_{Hadamard}(t)=\frac{1}{N}W(t)H_{N}$$

Therefore, the final response matrix $K_{Hadamard}(t)$ obtained using spatial encoding is in the same format as the FMC response matrix without spatial encoding, both containing $N^{2}$ echo signals. To continue with the example using N=2, the FMC response matrix without spatial encoding is as follows:(23)$$K_{normal}(t)=\left[\begin{array}{cc} r_{11}(t)+N(t) & r_{21}(t)+N(t)\\ r_{12}(t)+N(t) & r_{22}(t)+N(t) \end{array}\right]$$

The response matrix obtained using the Hadamard matrix excitation is as follows:(24)$$K_{Hadamard}(t)=\left[\begin{array}{cc} r_{11}(t)+\frac{N(t)}{2} & r_{21}(t)+\frac{N(t)}{2}\\ r_{12}(t)+\frac{N(t)}{2} & r_{22}(t)+\frac{N(t)}{2} \end{array}\right]$$

In the equation, $r_{ij}(t)\;$represents the useful signal, and N(t) represents the noise signal.

It is obvious that the noise in each item of the data obtained using Hadamard matrix excitation is lower. This method can increase the SNR ratio by $\sqrt{N}$ times at the same testing efficiency (the same number of transmissions as the TFM). Currently, common phased array probes have 64 or more elements. By using spatial encoding, it is possible to improve the signal-to-noise ratio by eight times or more, which is a very significant improvement.

### 3.3. The Combination of Time Coding and Spatial Coding

By now, time coding and spatial coding have been introduced. Time encoding increases the transmitted energy by increasing the pulse length, thereby improving the SNR. Spatial encoding, on the other hand, uses specific transmission patterns to suppress noise based on average denoising techniques, thus improving the SNR. Can these two approaches be combined to achieve the ultimate SNR improvement?

Of course, it is possible to combine time encoding and spatial encoding, as they are not inherently conflicting, and their definitions of encoding are not the same. The basic principle of spatial encoding is to use phased array probes where each element transmits a time-encoded waveform. The timing and phase of element transmissions follow Hadamard spatial encoding. Because spatiotemporal encoding combines the advantages of both approaches, it can improve the signal-to-noise ratio by 10log(ML) dB, where M is the number of elements in the phased array probe, and L is the length of the time encoding [93].

However, spatial encoding places higher demands on ultrasound devices, as it requires the capability to customize waveforms and independently control each channel of the phased array probe. As a result, it has significant limitations and is currently not widely used.

## 4. The Application of Coded Excitation in Various Fields of Ultrasonic Testing

Section 2 and Section 3 provide a detailed introduction to the basic principles and types of coded excitation, as well as various optimization methods. Additionally, we have conducted a theoretical analysis on the contributions of coded excitation to the SNR and imaging efficiency. As mentioned at the beginning of this paper, coded excitation originated from the radar field and has been introduced to the ultrasonic testing field for about fifty years. Due to its outstanding performance, it has consistently remained a topic of significant research interest; especially in scenarios where a significant improvement in the SNR is required, coded excitation is very useful. This section will explore the applications of encoding excitation in various domains of ultrasound detection, specifically focusing on industrial ultrasound and medical ultrasound.

### 4.1. The Application of Coded Excitation in Industrial Ultrasonic Testing

#### 4.1.1. Guided Wave Ultrasonic Testing

Ultrasonic testing typically uses bulk waves, or sound waves that propagate through a material’s interior. The propagation path is not affected by surface geometry, and the waves can penetrate through the thickness of a material. In contrast, guided wave testing is a special type of ultrasonic testing that uses sound waves that propagate along the surface or interface of a material. It is mainly used for the long-range testing of relatively long structures, covering larger areas that can detect defects such as corrosion and cracks within the structure [94,95].

However, guided wave testing is susceptible to environmental noise and material scattering noise during the inspection process. Additionally, the tested materials are often long, resulting in gradual signal attenuation during propagation, leading to a poor SNR. In such cases, the use of encoded excitation can significantly improve the effectiveness of the inspection.

Yuan Yang and Ping Wang proposed a guided wave method based on multi-element encoding to monitor fractures in steel rails [55]. The authors directly pointed out the existing issue that traditional single-pulse excitation results in lower energy, limiting the monitoring distance and damage resolution of the system. In practical scenarios, steel rails are often quite long, and if the length of the rail that can be detected each time the probe is deployed is limited, the overall efficiency of the inspection will be greatly reduced. Therefore, the authors designed a multi-element encoding excitation method that significantly increased the amplitude of the echo signal under different damage conditions.

Similar methods have also been proposed by Zeng Fan, who utilized a hybrid encoding approach to improve the SNR in the structural integrity testing of oil and gas pipelines [40], as depicted in Figure 19.

Mehmet K. Yücel and colleagues proposed a novel signal processing technique using maximum length sequences and linear frequency-modulated excitation signals [96]. This technique utilizes dispersion compensation and cross-correlation to significantly improve the SNR of guided wave responses. The theoretical findings were experimentally validated on aluminum rods, and Table 2 presents the SNR gains of the two encoding excitations mentioned by the authors.

It can be seen that coded excitation has made significant contributions in the field of ultrasonic guided waves. The main application targets are long materials such as steel rails, long pipes, and rods, where coded excitation can significantly improve the SNR, increase the echo amplitude, and extend the detection length.

#### 4.1.2. Air-Coupled Ultrasonic Testing

Air-coupled ultrasonic testing is a non-contact ultrasonic inspection technique used to detect defects in solid materials or extract material properties. Unlike traditional coupled ultrasonic techniques, air-coupled ultrasound utilizes air as the propagation medium, transmitting ultrasonic waves through transducers and receiving the echo signals [98].

The advantage of air-coupled ultrasonic testing lies in its non-contact nature, avoiding any potential damage to the tested material or transducer caused by direct contact. This also makes the testing process more convenient. However, it does have some disadvantages compared to water-coupled or direct-coupled ultrasound techniques. The main drawback is the relatively high propagation losses in air-coupled ultrasonics. Due to the significant difference in density and acoustic impedance between air and solid materials, a portion of the energy is reflected or scattered at the air–material interface, resulting in signal attenuation. Therefore, coded excitation techniques can be employed to increase the transmission energy and compensate for the attenuation.

T.H. Gan and D.A. Hutchins proposed a method for air-coupled ultrasonic imaging using a broadband acoustic transducer and pulse compression technique [47]. They employed the transducer to transmit a linearly chirped signal and conducted experiments on material thickness measurement and defect detection. The experimental setup diagram is shown in Figure 20. The experimental conclusion was that using encoding excitation can improve the SNR and resolution.

Honggang Li investigated the application of P4 phase-coded excitation in air-coupled ultrasonic testing and discussed how to select the parameters of the P4 code to achieve optimal pulse compression effects [61]. His experimental results demonstrated that compared to traditional pulse compression techniques, using a hybrid signal processing method can improve the SNR by 12.11 dB and enhance the temporal resolution by approximately 35%. Figure 21 shows the signal processing flowchart provided by the author.

Jianying Tang et al. used Barker code excitation for air-coupled the Lamb wave detection of blind holes on metal plates [51]. It can be directly observed from the pulse compression result that the SNR was significantly improved.

#### 4.1.3. Single-Probe Bulk Wave Testing

Compared to guided wave testing, ultrasonic bulk wave testing has a wider range of applications. It detects the defects and heterogeneity inside the material by utilizing the propagation properties of ultrasound in the material. In ultrasonic bulk wave testing, a single frequency or multiple frequencies of waves that pass through the tested object are usually used to detect the internal defects and corrosion of the material. Compared to ultrasonic guide wave testing, ultrasonic bulk wave testing is more suitable for the detection of local defects inside the material and has higher accuracy and detection depth. It is a common inspection technology in medical, material testing, and other fields.

Daniel A. Kiefer et al. used coded excitation signals to simultaneously measure the thickness and sound velocity of elastic plates [70]. They proposed that the use of a Wiener filter can improve axial resolution and measurement accuracy. Their experimental results show the use of the Wiener filter yields a narrower main lobe, indicating that this method can more accurately measure the thickness and sound velocity of plates of different materials and sizes with a relatively small relative error.

A. Rodríguez-Martínez et al. used optimized linearly chirped coded pulses combined with split-spectrum processing algorithms to detect defects in carbon fiber-reinforced composites and glass fiber-reinforced composites with significant acoustic attenuation [74]. The results are shown in Figure 22, where a noticeable improvement in the SNR can be observed.

#### 4.1.4. Phased Array Bulk Wave Testing

If phased array ultrasound is used, the SNR can be significantly improved by using Hadamard spatial encoding, as mentioned in Section 3.

Frederic Dupont-Marillia et al. conducted a study on the application of phased array ultrasound detection technology for inspecting forged steel blocks with a maximum depth of 1000 mm [98]. They compared the Hadamard spatial encoding method with two other commonly used methods: full matrix capture (FMC) and plane wave transmission (PW). In the end, they concluded that the Hadamard spatial encoding method has advantages in terms of detection accuracy and the SNR. The comparative results are shown in Figure 23.

Eduardo Lopez Villaverde et al. significantly improved the image quality by combining Hadamard coding transmission with a time-reversal operator denoising algorithm [45]. The results of detecting a welded joint compared to traditional methods indicate that Hadamard encoding can significantly improve the SNR.

### 4.2. The Application of Coded Excitation in Medical Ultrasonic Testing

#### 4.2.1. Guided Wave Ultrasonic Testing

Similar to the application of ultrasonic guided waves in industrial ultrasonics introduced in Section 4.1, waveguides can also be used in medicine to detect longer tissues, such as long bones.

Huilin Zhang et al. studied the use of encoded excitation to improve the effectiveness of ultrasonic guided waves in assessing long bone fractures [99]. The conventional excitation methods suffer from a low amplitude and SNR due to the high attenuation of the ultrasonic wave propagation in bone. However, the authors overcame this limitation by utilizing Barker code excitation and optimal binary code excitation. The measurement signals obtained from Barker code and optimal binary code excitation were decoded using Finite Impulse Response Least-Squares Inverse Filter (FIR-LSIF) and compared with the signal from Sinusoidal Pulse (SP) excitation. The experimental results demonstrated that encoded excitation was highly effective in improving the SNR for long bone guided wave testing.

#### 4.2.2. Phased Array Ultrasound Imaging Detection

Thanassis Misaridis and Jørgen Arendt Jensen investigated the application of linear frequency modulation signals and mismatched filters in medical phased array imaging to meet the high demands for the resolution and SNR in medical ultrasound [49]. The authors demonstrated that the encoded excitation method effectively improved the depth of imaging.

Emelina P. Vienneau et al. proposed an encoded excitation framework to improve the signal-to-noise ratio of transcranial ultrasound imaging without affecting the frame rate [34]. Their research showed that using a 65 bit encoding, in experimental transcranial scans on adults, the SNR gain could reach 17.91 ± 0.96 dB. The imaging results are shown in Figure 24.

A. Nowicki et al. utilized the Golay-coded ultrasound imaging technique to simulate the detection of the blood flow in smaller blood vessels [100]. Their experimental results demonstrated that using encoded excitation can provide higher-quality blood flow imaging compared to traditional short pulses. Additionally, it enabled the detection of the blood flow direction and suppressed static echo signals from outside the blood vessels. 

#### 4.2.3. Multi-Beam Parallel Transmission High Frame Rate Imaging Detection

Bahman Lashkari et al. studied the use of mismatched encoded excitation, which involves using excitation signals with high autocorrelation and low cross-correlation, to identify and separate signal sources at the receiver, enabling transmission from multiple elements simultaneously and allowing for the spatial decoding of transmitted signals [81]. This approach makes the imaging process equivalent to continuous transmission, greatly improving the efficiency of phased array synthetic aperture imaging. 

Bae-Hyung Kim and Tai-Kyong Song studied a method for simultaneous multi-beam transmission with multi-zone focusing using modulated orthogonal codes [82]. This method involves simultaneously transmitting M orthogonal Golay codes composed of M complementary sequences in multiple directions. Each complementary code is convolved with L different depth orthogonal frequency-swept signals. Finally, the L compressed frequency-swept signals are independently focused and merged, providing dynamic focused beams along M scan lines, where each beam has transmit focusing in multiple zones. This method achieves a high SNR and efficiency. The comparison with traditional ultrasonic testing is illustrated in Figure 25.

## 5. Conclusions

This paper provides a review of coded excitation in the field of ultrasound testing. Coded excitation is a technique originating from the radar field, and its remarkable properties have attracted interest from other domains, leading to its introduction into the field of ultrasound. Firstly, in Section 2, this article introduces the basic steps of ultrasound testing and explains how coded excitation has been improved in the transmission of ultrasound waves and signal processing analysis. With concise derivations, this article explains the important concept of matched filtering, i.e., pulse compression. Various forms of CE are shown in Table 3, such as linear frequency modulation (LFM) signals, biphasic coded signals such as Barker code signals and Golay code signals, as well as diverse coded signals such as P3 and P4 code signals, M-sequence signals and Gold sequence signals, and spatial encoding such as Hadamard spatial encoding. The combination of these encoding methods with pulse compression filters or other forms of decoding can significantly increase incident energy to improve the SNR and enhance detection depth. The advantages and disadvantages of various encoding methods are also summarized in the table above. In practical applications, appropriate selections should be made, and some encoding methods can be combined with optimization techniques to achieve an even better effect. In Section 4 of this paper, many application examples are provided to substantiate the practical feasibility of CE.

The paper categorizes CE into two classes: “time encoding” and “space encoding.” Each of these classes also has several subcategories, making the field of CE extensive. In addition to the various encoding types already discussed in this paper, there are other techniques such as Frank encoding in time encoding and S-sequence encoding in spatial encoding [101]. The extensive branches in this field also imply the potential for cross-fusion with many other technologies. We have noticed that some scholars have combined CE with signal processing methods such as compressive sensing [102] and wavelet transform [103], indicating that there will be more possibilities in the future. At the same time, we have noticed that CE has been widely applied and extensively researched in industrial ultrasonic testing and medical ultrasound imaging. Remarkable achievements have been made in guided wave testing and bulk wave testing. This reflects that CE is not confined to specific scenarios but rather is a versatile method in various fields.

Of course, CE is not without drawbacks. Long coded transmissions can result in large near-field blind zones and cause the excessive heating of the transducer. These issues have been addressed by researchers such as Julio Isla et al., who proposed interval coding to mitigate the near-field blind zone [104], and Mudabbir Tufail Bhatti et al., who suggested the use of a Capacitive Micro-Machined Ultrasonic Transducer (CMUT) to suppress the heating phenomenon [105]. In the future, more methods may emerge to overcome the limitations of CE [106]. We also note that CE imposes certain requirements on the device, especially in the case of LFM and spatial coding. Therefore, it is necessary to select the appropriate excitation method based on the capabilities of the instrument. In the future, as the performance of ultrasound devices continues to advance, the upper limit of CE will also be further elevated.

The following are some suggestions for future research based on the content of this paper. Firstly, in terms of theory, the principles of coded excitation should be fully integrated with the principles of ultrasound. Due to the diverse forms of ultrasonic detection, such as bulk waves and guided waves, as well as acousto-optics and magneto-acoustics not mentioned in this paper, attention should be paid to the mechanism of acoustic wave propagation to identify the differences between acoustics and the radar field of coded excitation, and to better integrate coded excitation and ultrasonic testing. Secondly, efforts should be made to develop higher-performance instruments. As mentioned earlier, different forms of coded excitation are suitable for different detection objects. If a device capable of freely adjusting and emitting waveforms could be developed, the flexibility of coded excitation detection would be greatly enhanced. Thirdly, efforts should be made to explore scenarios where traditional ultrasound detection methods face challenges, allowing coded excitation to demonstrate its usefulness.

## Figures and Tables

**Figure 1 sensors-24-02167-f001:**
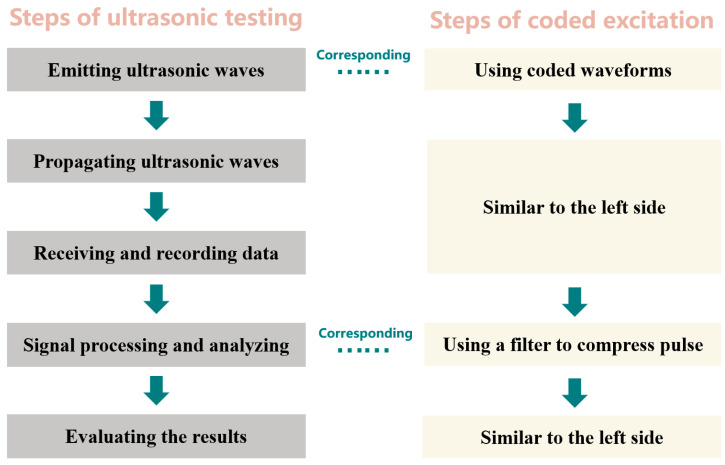
Steps of ultrasonic testing and coded excitation.

**Figure 2 sensors-24-02167-f002:**
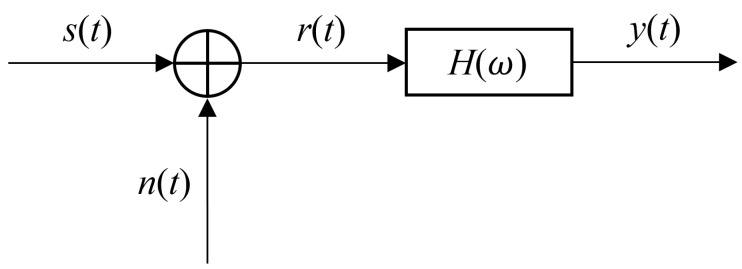
Flowchart of ultrasound echo signal processing.

**Figure 3 sensors-24-02167-f003:**
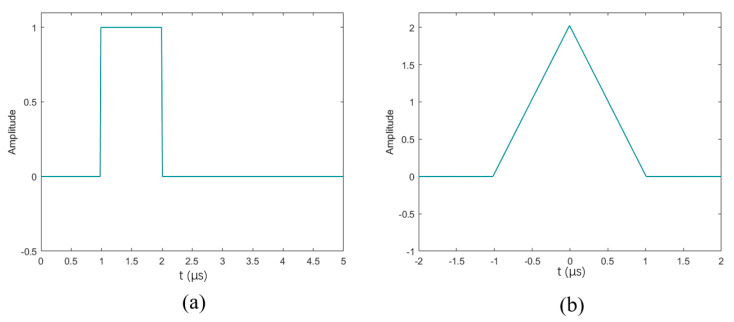
Simple pulse and its matched filtering output. (**a**) simple pulse; (**b**) matched filtering output of (**a**).

**Figure 4 sensors-24-02167-f004:**
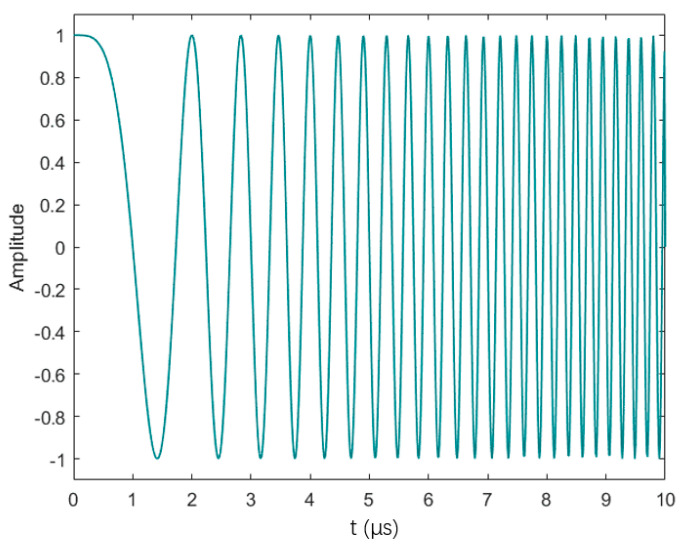
Linear frequency modulation signal.

**Figure 5 sensors-24-02167-f005:**
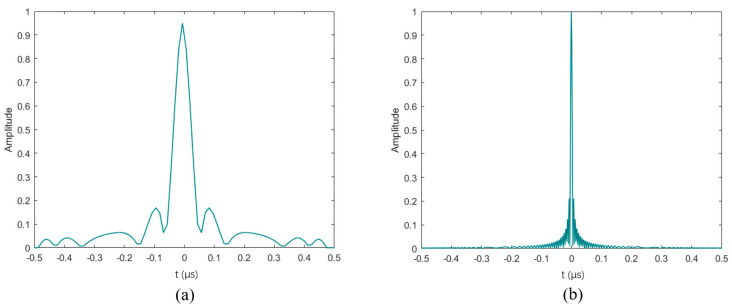
The matched filter output of LFM signal. (**a**) βτ = 10; (**b**) βτ = 100.

**Figure 6 sensors-24-02167-f006:**
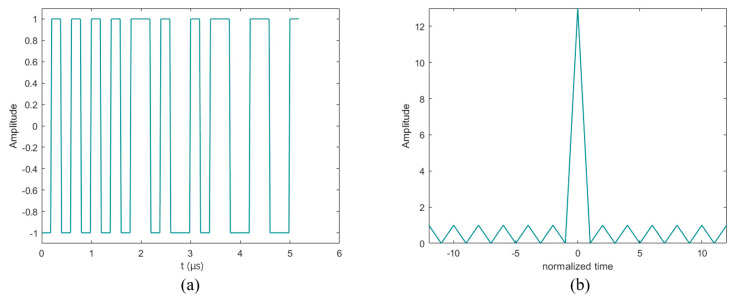
The 13-bit Barker code waveform and the matched filter output. (**a**) The 13-bit Barker code waveform; (**b**) the 13-bit Barker code matched filter output.

**Figure 7 sensors-24-02167-f007:**
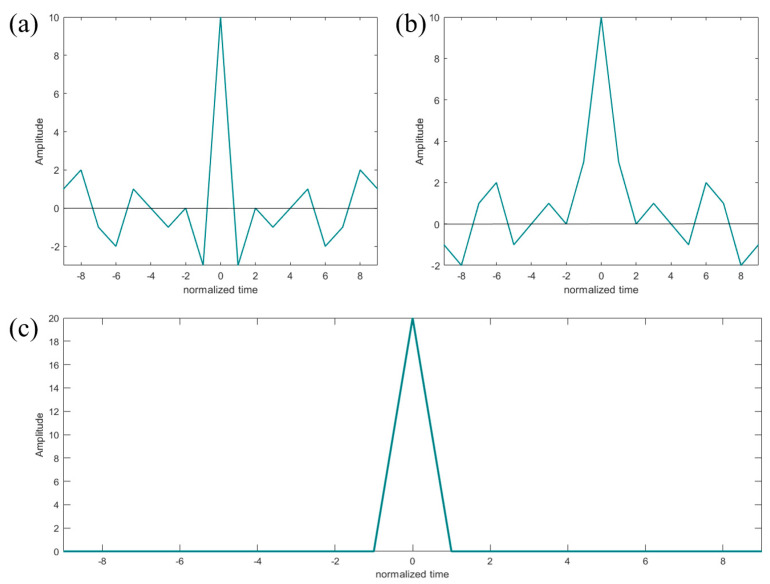
Example of matched filter output and their sum for a 10-bit Golay complementary pair. (**a**,**b**) matched filter output of 10-bit Golay complementary pair; (**c**) sum of (**a**,**b**).

**Figure 8 sensors-24-02167-f008:**
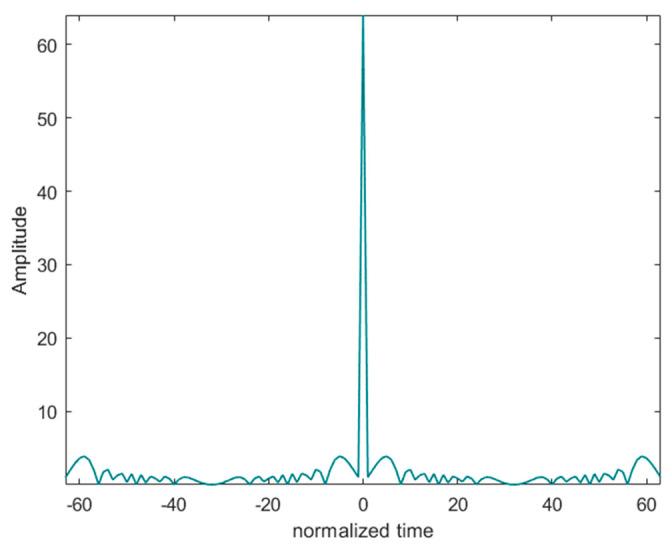
Matching filter output of 64 bit P3 code.

**Figure 9 sensors-24-02167-f009:**
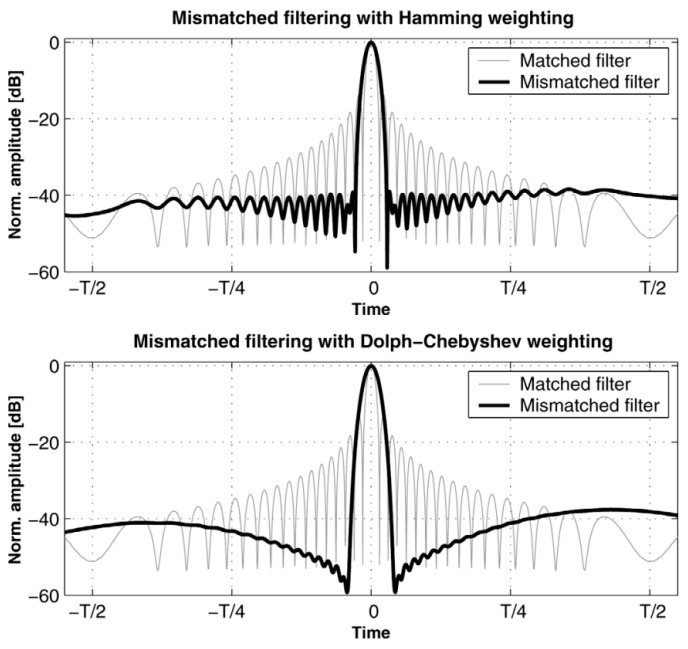
Compression outputs for two mismatched filters based on time weighting with Hamming (upper graph) and Dolph–Chebyshev windowing (lower graph) [27].

**Figure 10 sensors-24-02167-f010:**
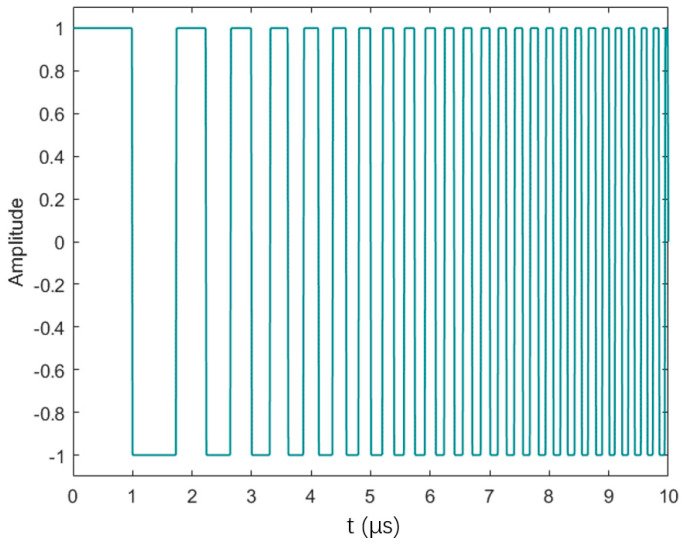
Rectangular linear frequency modulation signal.

**Figure 11 sensors-24-02167-f011:**
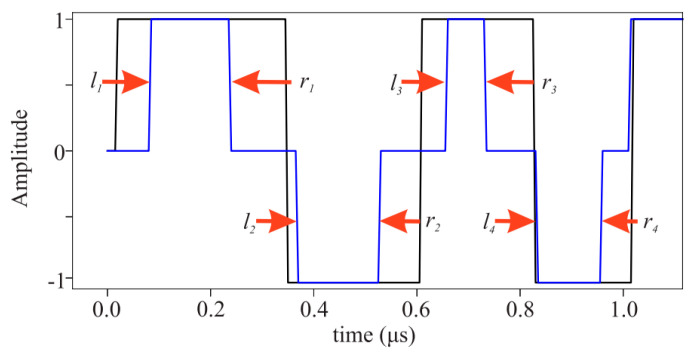
An example of width reduction at a particular iteration at step 2 of the optimization algorithm. RNLFM chirp seed (black), APWP prototype (blue), and red arrows that show the pulse reduction direction at stages 2 and 2b (left and right reduction, respectively) [74].

**Figure 12 sensors-24-02167-f012:**
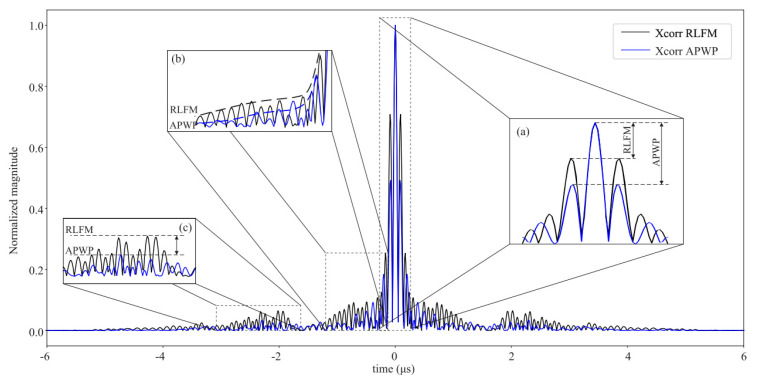
Cross-correlation function for RLEM black and optimized APWP blue signals’ comparison of (**a**) main-to-secondary lobe level, (**b**) sidelobes’ energy leakage, and (**c**) width and height of lateral lobes [74].

**Figure 13 sensors-24-02167-f013:**
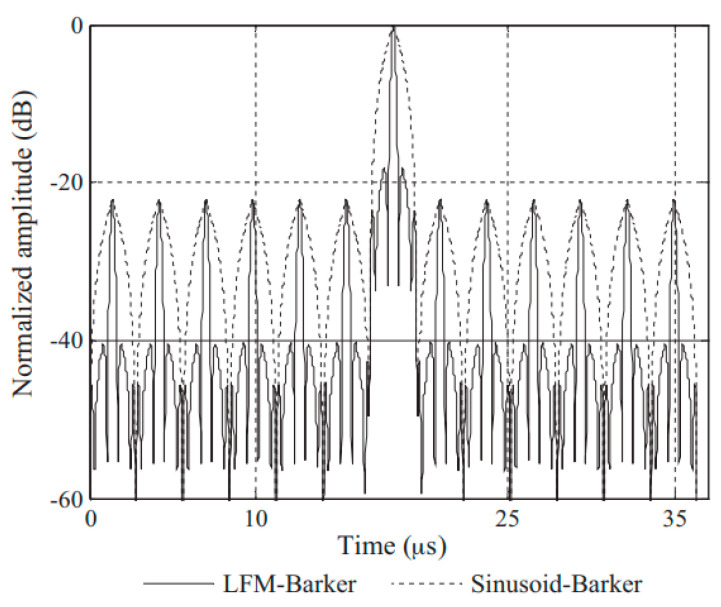
The autocorrelation function envelopes of the LFM–Barker and sinusoid–Barker coded signals [79].

**Figure 14 sensors-24-02167-f014:**
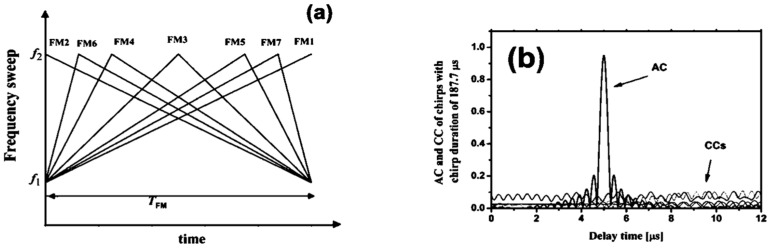
(**a**) Seven mismatched LFM signals. (**b**) Autocorrelation results of FM4 signal and cross-correlation results of FM2, FM3, and FM5 signals [81].

**Figure 15 sensors-24-02167-f015:**
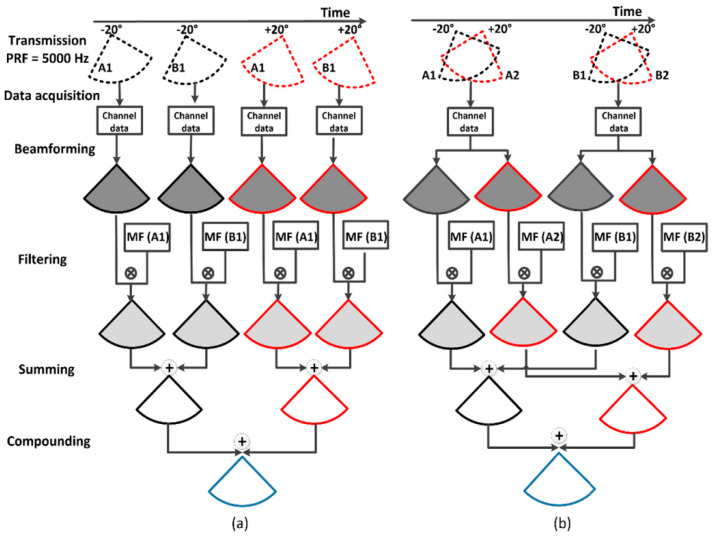
Imaging sequences for (**a**) using regular Golay pairs. (**b**) Using orthogonal Golay pairs [84].

**Figure 16 sensors-24-02167-f016:**
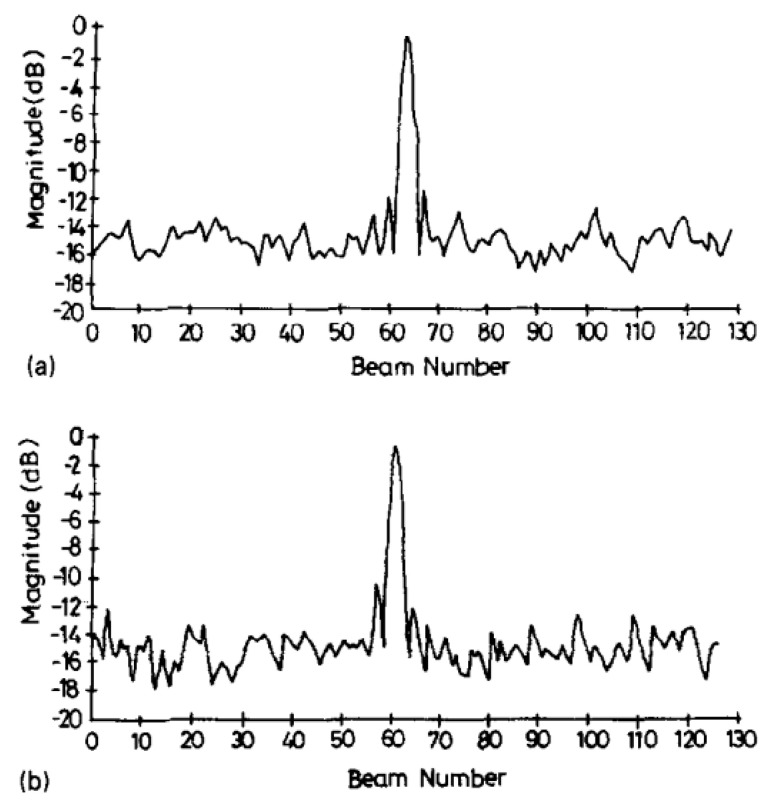
Transmit beam profiles obtained by using 8191 sequences and (**a**) correlator and (**b**) Wiener filter [87].

**Figure 17 sensors-24-02167-f017:**
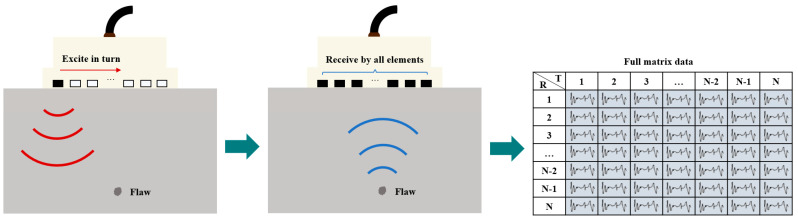
The process of full matrix capture (FMC). The red lines represent the incident sound wave, and the blue lines represent the echo wave.

**Figure 18 sensors-24-02167-f018:**
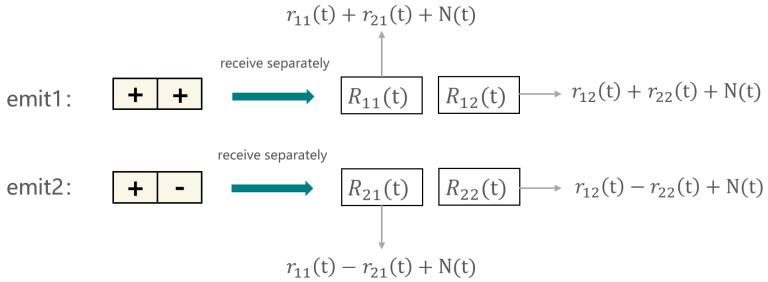
Hadamard encoded transmission and retrieval signals.

**Figure 19 sensors-24-02167-f019:**
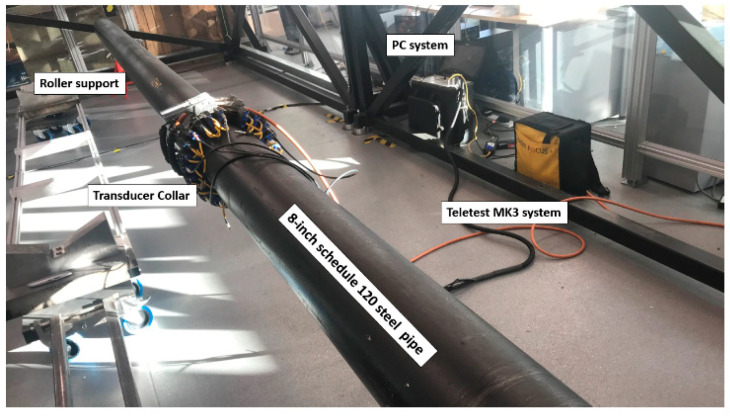
Hadamard encoded transmission and retrieval signals [40].

**Figure 20 sensors-24-02167-f020:**
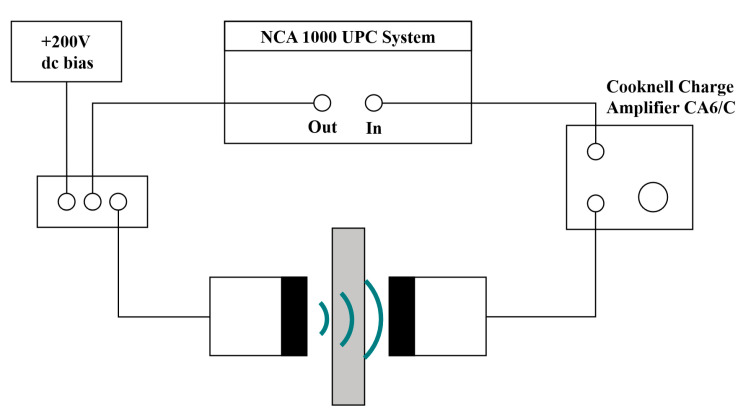
Experimental arrangement for through-transmission testing using pulse compression.

**Figure 21 sensors-24-02167-f021:**
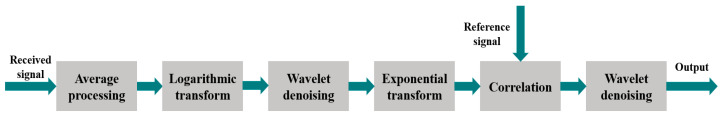
Signal processing block diagram.

**Figure 22 sensors-24-02167-f022:**
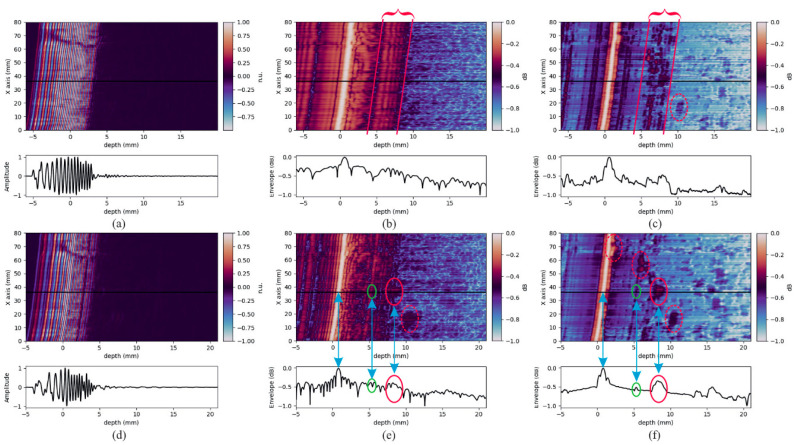
Results for carbon–aluminum composite with 5 MHz transducer. In each subfigure, upper graphic shows B-scan and lower graphic A-scan at particular position (horizontal black line in B-scans). (**a**) Raw signal for RLFM, (**b**) compressed signal before SSP for RLFM, (**c**) result after SSP for RLFM, (**d**) raw signal for APWP, (**e**) compressed signal before SSP for APWP, and (**f**) result after SSP for APWP [74].

**Figure 23 sensors-24-02167-f023:**
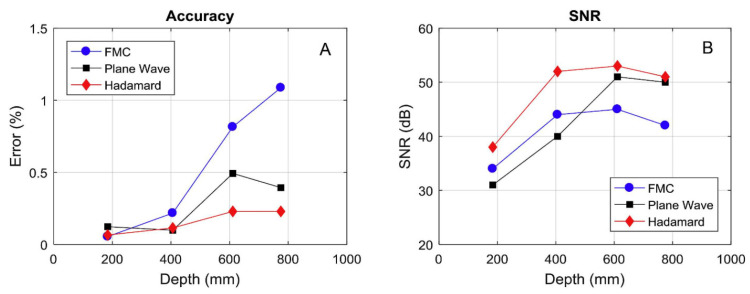
Experimental notch position accuracy (**A**) and signal-to-noise ratio (**B**) for phased array imaging using FMC, PW, and Hadamard matrix transmission sequences [98].

**Figure 24 sensors-24-02167-f024:**
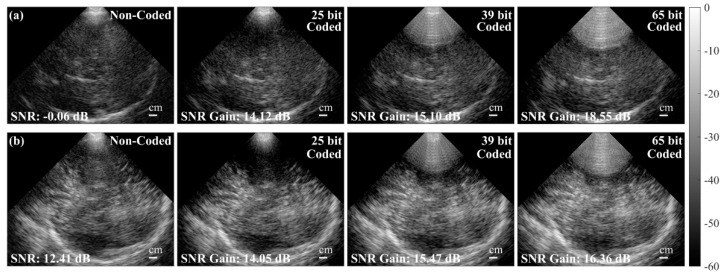
In vivo transcranial B-Mode imaging with and without coded excitation in two subjects. (**a**) Top row: low image quality case (Subject 3, a 50-year-old Caucasian female). (**b**) Bottom row: high image quality case (Subject 5, a 24-year-old Caucasian male). In both cases, the SNR gain increases with increasing code length. The greatest gain is achieved with the 65 bit code in the low image quality case. The 39 bit and 65 bit codes show the same nearfield artifact from transmitting and receiving at the same time (the dead zone) [34].

**Figure 25 sensors-24-02167-f025:**
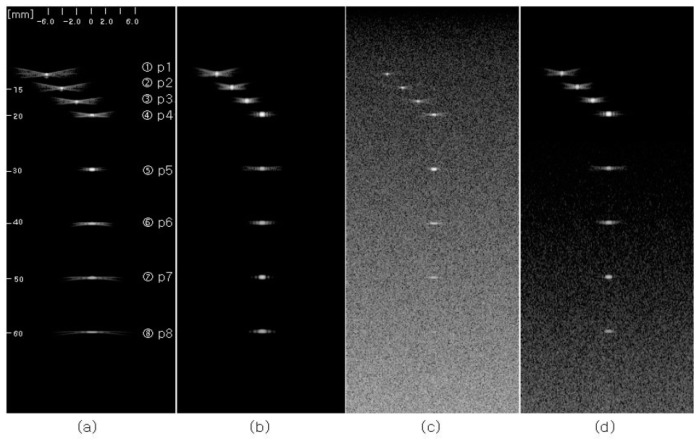
B-scan images (dynamic range: 60 dB) of point targets using (**a**) fixed transmit focusing at 30 mm with the conventional pulse-echo method and (**b**) the MB-STMZ focusing with frequency range [3.5 MHz 8.2 MHz], focused at 20 mm with frequency range [11.5 MHz 6.8 MHz], focused at 50 mm. Panels (**c**) and (**d**) show the images for the same methods used for panels (**a**) and (**b**), respectively, when additive white Gaussian noises are added so that the PSNR is 25 dB [82].

**Table 1 sensors-24-02167-t001:** Barker code.

Code Length	+/− Format	Peak Sidelobe (dB)
2	+1 −1	−6.0
2	+1 +1	−6.0
3	+1 +1 −1	−9.5
4	+1 +1 −1 +1	−12.0
4	+1 +1 +1 −1	−12.0
5	+1 +1 +1 −1 +1	−14.0
7	+1 +1 +1 −1 −1 +1 −1	−16.9
11	+1 +1 +1 −1 −1 −1 +1 −1 −1 +1 −1	−20.8
13	+1 +1 +1 +1 +1 −1 −1 +1 +1 −1 +1 −1 +1	−22.3

**Table 2 sensors-24-02167-t002:** SNR gains of the two encoding excitations [97].

Signal	Mode	Raw (dB)	Result (dB)	Increase (dB)
LFM	Longitudinal Flexural	19.0 13.3	26.6 24.4	7.6 11.1
MLS	Longitudinal Flexural	22.5 18.6	28.0 23.4	5.5 4.8

**Table 3 sensors-24-02167-t003:** Summary of various coded excitation methods.

Coded Excitation Name	Type	Advantage	Disadvantage
Without coding (simple pulse)	/	Easy to launch Low device requirements	SNR coupled with resolution
Linear frequency modulation (LFM)	Time coding	Has been extensively studied The time width and bandwidth of the signal can be freely adjusted There are many ways to optimize it	Limited by device performance
Barker code	Time coding	Relatively good autocorrelation performance (low sidelobes) The code length is moderate Easy to launch with minimal device limitations There are several ways to optimize it	The maximum code length is 13 bits, so the upper limit of the SNR gain is relatively low
Golay code	Time coding	Can completely eliminate sidelobes (eliminating artifacts) No limit on code length There are several ways to optimize it	The same position needs to be emitted twice, so the detection efficiency is relatively low
P3, P4 code	Time coding	With lower sidelobe level and greater Doppler tolerance	The existence of sidelobes implies the presence of artifacts
M-sequence	Time coding	Easy to generate Can be utilized for efficient scanning with multi-beam parallel transmission by leveraging its “encryption” property	The sidelobe suppression is not as good as the Barker code and Golay code
Gold sequence	Time coding	The quantity is much greater than the M-sequence Can be utilized for efficient scanning with multi-beam parallel transmission by leveraging its “encryption” property	The autocorrelation is not as good as the M-sequence
Hadamard encoding	Spatial coding	Combined with the TFM method to achieve a high SNR and resolution	Must be achieved by using phased array ultrasound probes
Combination of time coding and spatial coding	Time coding and spatial coding	Can achieve the ultimate SNR and resolution	Severely limited by the device

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
