# Peer review of "Coded Excitation for Ultrasonic Testing: A Review"

_sensors, 2024, doi:10.3390/s24072167_

Round 1
Reviewer 1 Report
Comments and Suggestions for Authors INTRODUCTIONThis paper is a revision paper that aims to present various coded excitation techniques for ultrasound pulses and the improvement achieved by different techniques. The paper is well-structured, with Section 2 detailing the principles of code excitation, providing examples and discussing main optimization techniques (prior to transmission but also in post-processing). Section 3 describes coded excitation for improving detection and spatial encoding. In Section 4, various applications (guided waves, airborne ultrasounds, phased array, medical imaging…) utilizing different CE techniques are also discussed. Finally, the main conclusions are presented.
The paper contains sufficient references and thoroughly explains the fundamental aspects of coded excitation, making it easy to comprehend for readers. While no scientific breakthroughs are achieved as it is a review paper, this work serves as a comprehensive summary.
SOME ASPECTS THAT SHOULD BE CLARIFIED OR CHECKED
· Section 6 is not included, although it is presented at the end of section 1. The reviewer kindly suggests including this section or removing the sentence “Finally, in Section 6, the future research directions for coded excitation were summarized and discussed.” from Section 1
· Equation (4) and (6). The parameters \beta and \tau control the bandwidth (\beta) and the duration ($\tau$) of the pulse of the chirp signal. It is a bit confusing that \beta and \tau appear as fractions ($\frac{\beta}{\tau}$) in equations 4, 5, and 6, but after that, the authors define a value that is $\beta \times \tau$. The reviewer understands this definition, but some explanation should be included.
· Additionally, the authors use two values of $\beta\tau$: 10 and 100. Considering a 10us duration signal (Figure 4), $\beta$ should be 1MHz and 10MHZ bandwidth. But analysing Figure 5, it seems that $\tau$ should be 1 $\mu s$ and $\beta$ 10MHz and 100MHz respectively. The reviewer kindly suggests including some sentences clarifying these aspects: $\beta \times \tau$ and the values of $\tau$ and $\beta$.
· Figure 5 corresponds to $\beta\tau$ 10 and 100 respectively, but it seems that \tau must be 1\mus. This should be clarified because Figure 4 induces misleading.
· Figures 6b, 7, and 8 correspond to the matched filter output of different signals. The x-labels say “normalized time”. What does “normalized time” mean? The authors sometimes use “time (\mu s”) for the output of the matched filter. Meanwhile, other times, they use “normalized time”. They should use the same units or clarify the differences between them.
· Figure 9. What does T represent?
· Equations 15 and 16 are confusing as the compensating function A(t) is defined in the time domain; meanwhile, in Equation 16, authors try to work in the frequency domain.
· Section 4 described code pulse codification in different fields and applications. Quantitative values are provided in some applications, while qualitative values are provided in others.
· The spatial encoding is adequately described, but Figure 18 is a bit confusing despite this effort. Maybe the authors could try to improve it. As a kindly suggestion, it might help if R12 and R22 were described completely as R11 and R21 are.
· Using $f$ or $\omega$ to represent the frequency domain is valid, but the reviewer recommends using one and not mixing them through the work. Equation 13 uses f. Meanwhile, the authors use \omega before this equation.
MISSPELINGS
· Equation (3). The parenthesis should include both terms of the fraction
· Reference 14 is part written in red.
· Section 6 (Future work) is named at the end of section 1, but it is not included in the paper.
· Figure 2. The multiplication symbol should be an addition symbol. As the figure is, it seems that r(t) = s(t)·n(t).
· Figure 4, 5. X-Labels are confusing. They are expressed as t /\mus. The reviewer guesses that the authors want to highlight the units (which is very correct), but even he/she knows the units should be written between brackets or parenthesis.
· Page 11: “Assuming that the spectrum of the coded pulse x(t) is x(f)” it should be “Assuming that the spectrum of the coded pulse x(t) is X(f)”
· Figure 11. In the caption, “algorithm” word appears written in red
· Figure 14 is captured from other work, and they are referenced. The quality of the figure should be improved.
· Table 3. The reviewer is not a native English speaker, but some sentences seem to be not properly written: some sentences begin with a capital letter (as expected), others with lowercase letters, and others seem to be missing the sentence's subject. Please, could the authors revise it?
CONCLUSIONS
The paper is a good revision paper and contains the main aspects of code excitation. As drawbacks of this work, we could point out that some sections are widely described (as spatial coding). Meanwhile, others are not so much.
Some corrections should be carried out.
Author Response
Thank you for your review comments. We have responded to each comment in the attachment one by one.

Reviewer 2 Report
Comments and Suggestions for Authors
The manuscript titled "Coded Excitation for Ultrasonic Testing: A Review" provides a comprehensive overview of the principles, applications, and optimization techniques of coded excitation in ultrasonic testing. The authors have effectively highlighted the importance of achieving a balance between signal-to-noise ratio and imaging efficiency in ultrasonic testing and discussed how coded excitation can address this challenge. The paper is well-structured and provides valuable insights into the application of coded excitation in different areas of UT, including industrial bulk wave single probe detection, industrial guided wave detection, industrial bulk wave phased array detection, and medical phased array imaging. Overall, this review manuscript is well-researched and contributes to the existing literature on ultrasonic testing techniques.
1) The introduction provides a clear and concise overview of NDT and its significance in various industries. It effectively sets the context for the subsequent discussion on ultrasonic testing and the need for improved SNR and imaging efficiency. However, it would be beneficial to provide a brief explanation of the limitations of conventional short pulse excitation methods before introducing the concept of CE.
2) The manuscript follows a logical structure, with each section addressing specific aspects of CE in ultrasonic testing. The division of the paper into sections dedicated to the basic principles, optimization, and applications in different areas of UT is appropriate and facilitates easy comprehension. Additionally, the inclusion of a comprehensive table detailing various types of coded excitation and their characteristics in the final section is highly informative.
3) The authors have provided a thorough review of the fundamentals of CE, including signal modulation, transmission, reception, pulse compression, and optimization techniques. The discussion on the application in different areas of ultrasonic testing is comprehensive and covers both industrial and medical domains. The inclusion of references to relevant studies and the description of specific use cases further enhance the credibility of the review. However, it would be valuable to include more recent references, particularly in the discussion of advancements and future directions in CE.
4) The manuscript is well-written and organized, with clear and concise language used throughout. The technical concepts are explained effectively, making the content accessible to readers with varying levels of expertise in UT. However, in some instances, the manuscript assumes a certain level of familiarity with the topic, particularly in the explanation of CE principles. It would be beneficial to provide additional clarifications or examples to ensure comprehensive understanding.
5) The conclusion provides a concise summary of the main findings and highlights the advantages of CE in UT. The discussion on future research directions for coded excitation is insightful and encourages further exploration in this field. However, it would be beneficial to provide more specific suggestions or areas of focus for future studies to guide researchers interested in advancing CE techniques in ultrasonic testing.
Author Response

(The authors gave the same response as above.)
